# BINCTX: MULTI-MODAL REPRESENTATION LEARNING FOR ROBUST ANDROID APP BEHAVIOR DETECTION

## ABSTRACT

Mobile app markets host millions of apps, yet undesired behaviors (e.g., disruptive ads, illegal redirection, payment deception) remain hard to catch because they often do *not* rely on permission-protected APIs and can be easily camouflaged via UI or metadata edits. We present BINCTX, a learning approach that builds *multi-modal* representations of an app from (i) a global bytecode-as-image view that captures code-level semantics and family-style patterns, (ii) a contextual view (manifested actions, components, declared permissions, URL/IP constants) indicating how behaviors are triggered, and (iii) a third-party-library usage view summarizing invocation frequencies along inter-component call paths. The three views are embedded and fused to train a contextual-aware classifier. On real-world malware and benign apps, BINCTX attains a macro $F_1$ of $94.73\%$, outperforming strong baselines by at least $14.92\%$. It remains robust under commercial obfuscation (F1 $84\%$ post-obfuscation) and is more resistant to adversarial samples than state-of-the-art bytecode-only systems.

## 1 INTRODUCTION

Mobile applications (apps) have become integral to daily life. While apps bring convenience, app quality and compliance remain a concern across markets. The number of malicious apps (malware) continues to grow with more sophisticated evasion techniques. At the same time, driven by commercial incentives, many apps exhibit undesired behaviors (e.g., ad disruption or payment deception) that degrade user experience or violate market policies Hu et al. (2021).

To mitigate these issues, app markets publish developer policies Google (2021) and deploy vetting pipelines such as Google Play Protect (GPP) Google (2022b), which combine static and dynamic analysis and involve human review for suspicious cases. However, studies show that a substantial portion of potentially harmful apps (PHAs) still evade detection Riley (2019); McLaughlin et al. (2017); Sun et al. (2019); Arp et al. (2014); Türker & Can (2019), partly due to evolving malware tactics Tam et al. (2017). Signature-based defenses struggle with the proliferation of variants, and learning-based approaches using permissions, API calls, opcodes, or XML signals Arp et al. (2014); Aafer et al. (2013); Xu et al. (2018); McLaughlin et al. (2017); Kim et al. (2019) also face practical limitations.

We summarize four challenges for detecting both malware and undesired behaviors:

1. Limited coverage of permission-protected APIs. Many detectors emphasize permission-protected APIs Xi et al. (2019); Zhang et al. (2014); Karbab et al. (2017); Peiravian & Zhu (2013); Onwuzurike et al. (2019); Ma et al. (2019); Shen et al. (2017); Hou et al. (2017), yet not all malicious or undesired behaviors rely on such APIs. For example, aggressive advertising or payment deception via third-party SDKs may not require dangerous permissions Demetriou et al. (2016); Hu et al. (2021); Huang et al. (2014). Modeling beyond permission-protected calls is necessary.

2. Insufficient modeling of behavioral context. Undesired or malicious behaviors are often triggered via background services or system events Yang et al. (2015); Xi et al. (2019). Prior work on code summarization Alon et al. (2019); Xu et al. (2019); Allamanis et al. (2016); Hu et al. (2018); Iyer et al. (2016); LeClair et al. (2020); Zhang et al. (2019a) focuses on implementation structure and provides limited coverage of when and how behaviors are activated. Signals like UI layouts or coarse metadata may be easy to modify and loosely tied to code semantics Xu et al. (2018).

3. Weak commonality among undesired behaviors. Many policy-violating behaviors (e.g., ad disruption) do not show strong family-level regularities, and are closely related to third-party library usage patterns Hu et al. (2021); Wang et al. (2022); Wang & Guo (2017); Martín et al. (2017); Crussell et al. (2014); Son et al. (2016); Demetriou et al. (2016); Grace et al. (2012); Shao et al. (2018); Jin et al. (2021). Detectors that overlook SDK usage and its context struggle to generalize.

4. Evasion via obfuscation and adversarial manipulation. Code obfuscation jia (2023); Maiorca et al. (2015) and adversarial manipulations Li & Li (2020); Huang et al. (2021); Chen et al. (2017); Li et al. (2021) can distort specific opcode or manifest cues while preserving behavior, which hurts detectors that rely on a single feature type.

We present BINCTX, which combines code-level signals and behavioral context to detect malware and undesired behaviors. The approach jointly uses: (i) a global bytecode representation that maps the entire DEX file to an RGB image and extracts a CNN embedding (DenseNet) to capture code-level regularities Sun et al. (2019); Kang et al. (2020); Huang et al. (2017); He et al. (2016); Simonyan & Zisserman (2015); Tan & Le (2019); (ii) a contextual representation from AndroidManifest.xml (declared components, intent actions, permissions) and from code/resources (URL and IP constants), which reflects triggers and destinations; and (iii) a third-party library usage representation based on inter-component call graphs (ICCGs), where call-path counting summarizes how SDK APIs are exercised. These representations are concatenated and fed to a multi-layer perceptron classifier. When one signal is perturbed (for example, dead code insertion), the remaining signals (contextual triggers and SDK usage) still constrain behavior and improve robustness.

Our main contributions are as follows. (1) We develop BINCTX, a combined representation for app behavior that includes a global bytecode embedding, contextual features, and third-party SDK usage patterns, enabling the detection of both malware and undesired behaviors (ad disruption, illegal redirection, payment deception). (2) We implement feature extractors for DEX-to-image embedding, manifest and code/resource parsing, and ICCG-based SDK path counting, and train a compact classifier over the concatenated features. (3) We evaluate on real-world malware and benign apps as well as labeled undesired behaviors. BINCTX achieves an average $F_1$ of $94.73\%$, improving over prior approaches by at least $14.92\%$. Ablations and permutation importance show that all three feature groups contribute, and the combined representation is more resistant to obfuscation and adversarial manipulations than baselines. (4) We provide implementation and datasets to facilitate follow-up work BINCTX (2025).

## 2 BACKGROUND AND MOTIVATION

**Android bytecode (DEX).** An Android APK contains one or more `classes.dex` files[1]. Each DEX stores compiled program elements (activities, classes, methods, code) executed by the Android Runtime. The file consists of three major sections: a *header* (magic/version, checksums, file size, and offsets/sizes of other regions), an *index* area with identifier lists (strings, types, prototypes, fields, methods), and a *data* area containing class definitions, code items, and other payloads. These structures are byte-addressable and can be processed without decompilation.

**Image representation of bytecode.** Following prior work Sun et al. (2019), we convert DEX to an RGB image by reading bytes as a hex stream, mapping each 6 hex digits to one pixel (3 bytes), filling row-major, and padding with $(0, 0, 0)$ when needed. This preserves local byte neighborhoods and avoids brittleness from incomplete decompilation. In practice we resize/crop to a fixed input (e.g., $300 \times 300$) for CNN backbones.

**Motivation.** Rendered as images, samples from the same malware family present similar textures, while different families show distinct patterns (Figure 1). This provides a global code-level cue resilient to identifier changes. For undesired behaviors (e.g., ad disruption, payment deception) that may not share strong bytecode-level regularities, contextual signals such as manifest-declared components/actions and URL/IP constants become important indicators of triggers and destinations, motivating the combination used in our approach.

---

[1]Sometimes multiple files, e.g., `classes2.dex`.

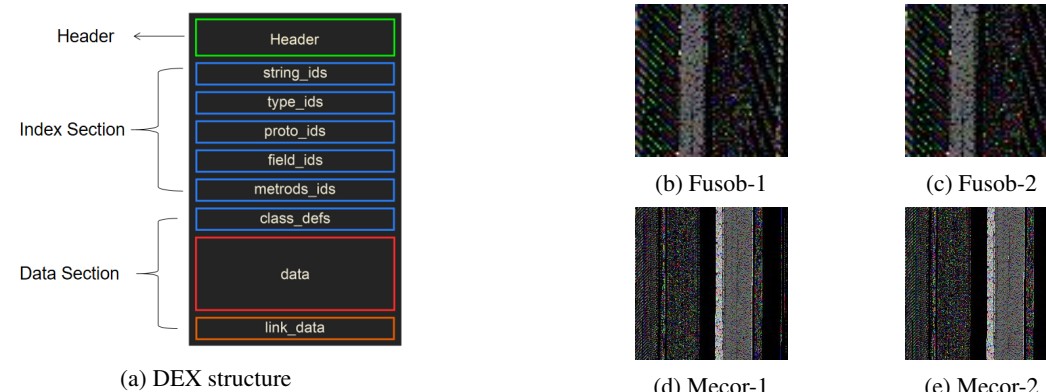

(a) DEX structure

(b) Fusob-1

(c) Fusob-2

(d) Mecor-1

(e) Mecor-2

Figure 1: (a) Structure of an Android bytecode file (`classes.dex`). (b)–(c) Fusob (ransomware) samples: similar fine-grained diagonal textures and stripe clusters, illustrating strong intra-family regularity. (d)–(e) Mecor (potentially unwanted app (PUA)) samples: broader high-contrast vertical bands with fewer diagonal artifacts, distinct from Fusob yet consistent within the family.

## 3 APPROACH

### 3.1 OVERVIEW

Figure 2 shows the overview of BINCTX. Our methodology consists of two major phases: first, constructing a multi-modal representation of each app through three parallel feature extraction modules, and second, training a neural network to classify app behavior based on this fused representation. The feature extraction phase contains three modules, each taking an Android APK file as input: (1) a bytecode representation extraction module that outputs a visual, image-based representation of the app's code; (2) a contextual information extraction module that analyzes metadata and code to extract explicit behavioral triggers and endpoints; and (3) a third-party library extraction module that retrieves quantitative usage patterns of common SDKs. After feature extraction, the resulting feature vectors are used as input to train the contextual-aware classification model, which outputs the final predicted label for each app.

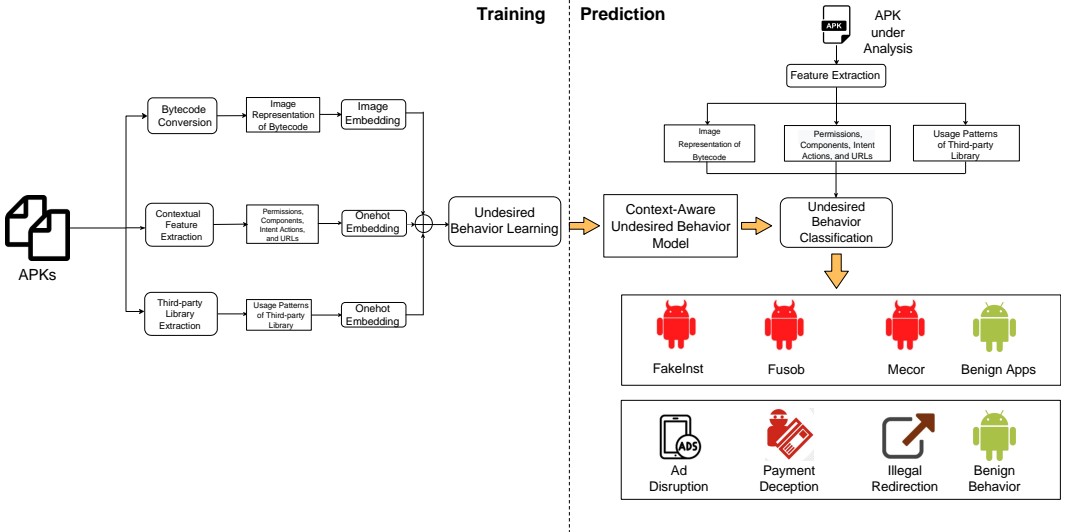

Figure 2: Overview of the BINCTX framework.

### 3.2 BYTECODE REPRESENTATION EXTRACTION

This module takes Android APK files as input and outputs the converted image representation of their bytecode in RGB format. As it is difficult to determine which specific part of the code triggers an

app's undesired behaviors, we model the app's behavior as a whole. While call graphs or program dependency graphs can represent an overview of app behavior, building them accurately and efficiently for Android is challenging due to complex life cycle events, system events, multi-threading, and third-party library dependencies Arzt et al. (2014); Rountev & Yan (2014). Alternatively, inspired by recent works Kang et al. (2020); Sun et al. (2019), an efficient yet effective way to model app behavior is to use the image representation of its bytecode. We treat the entire bytecode as a sequence of bytes and convert it into an image. This conversion is not only efficient but also reveals similarities in app behaviors through image patterns, which can be effectively processed by mature image recognition techniques.

**Bytecode Conversion.** To convert a bytecode file to an image, BINCTX directly reads the file as a sequence of hexadecimal numbers. Three consecutive bytes (six hexadecimal digits) are then interpreted as the R, G, and B values of a single pixel. For example, each Android bytecode file must start with a magic number `6465780a30334500`, which is converted into the pixels $(100, 101, 120), (10, 48, 51), \ldots$. These pixels are arranged from the top-left to the bottom-right corner to form the image. We use $(0, 0, 0)$ to pad any remaining space to ensure a uniform image size.

**Image Embedding.** We use a pre-trained DenseNet Huang et al. (2017) model to transform the bytecode image into a dense vector embedding. DenseNet's architecture contains multiple "dense blocks," where the input of the $l$-th layer is a concatenation of the outputs from all preceding layers ($X_l = f([H_1, H_2, ..., H_{l-1}])$). This structure encourages feature reuse and has proven highly effective at capturing the hierarchical patterns in visual data, which we hypothesize translates to discovering compositional patterns in the code's "texture." In our approach, we remove the final classification layer of DenseNet and use the output of the last transition layer as the bytecode image embedding, denoted as $f_{\text{bin}}$.

## 3.3 Contextual Information Extraction

As described in Section 1, undesired behaviors are typically enabled in the background or triggered by system events. Thus, we extract two types of contextual features: (1) declared components, permissions, and intent actions from the `AndroidManifest.xml` file, and (2) network address constants from the code.

Specifically, BINCTX analyzes the `AndroidManifest.xml` file to extract its declared permissions and components (*Activity*, *Service*, *Content Provider*, and *Broadcast Receiver*). For each component, we also extract the action names within its `intent-filter` section, as these describe the operations it can perform. Next, BINCTX extracts network address constants (URLs and IPs) from the code. We use Soot Vallee-Rai et al. (2000) to decode the APK files and then iterate through each instruction to locate assignment statements that assign URLs or IPs to variables. We also scan the string resource file (`strings.xml`) to find additional network constants.

**Feature Embedding.** We build a global vocabulary of size $V$ from all unique features (permissions, components, actions, network addresses) observed in the training data. Each application is then represented by a $V$-dimensional binary vector, $f_{\text{cxt}} \in \{0, 1\}^V$, where each dimension indicates the presence (1) or absence (0) of a specific feature.

## 3.4 Third Party Library Extraction

To represent the usage patterns of third-party libraries, BINCTX identifies which libraries are used and quantifies their API invocation frequency by counting the number of call paths leading to their APIs. This requires the construction of a comprehensive Inter-Component Call Graph (ICCG).

While tools like FlowDroid Arzt et al. (2014) can build call graphs for Java, they often fail to capture the unique characteristics of Android apps. To address this, we first leverage FlowDroid to build a static call graph from standard invoke statements. We then expand this graph with edges representing implicit calling relationships common in Android (e.g., UI handlers, lifecycle methods). Furthermore, we integrate Inter-Component Communication (ICC) method calls using the ic3 tool Octeau et al. (2015), which joins the call graphs of otherwise disconnected components. Finally, a dummy main method node connects all components to form the complete ICCG.

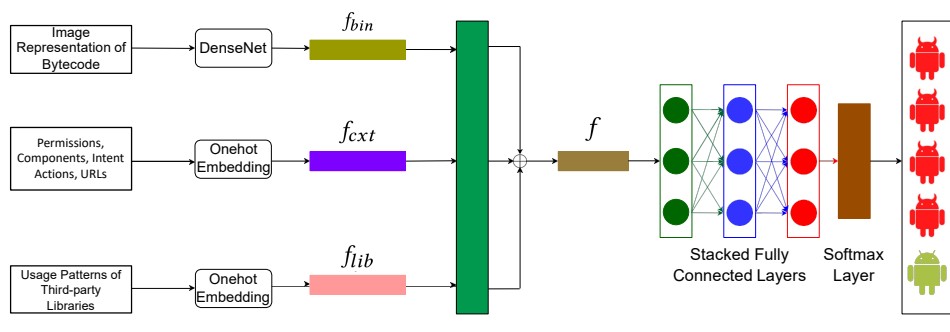

Figure 3: Overview of Contextual-Aware Undesired Behavior Model

Based on these ICCGs, we count the call paths to a curated list of widely used third-party libraries for services like advertising, maps, and payments Google (2022a); PayPal (2021); Alibaba (2021). Using `networkx` NetworkX (2022), we apply a depth-first search to extract all paths leading to the APIs of these libraries, removing any self-loops.

**Feature Embedding.** For a set of identified third-party libraries $L = \{l_1, l_2, ..., l_k\}$, we denote the number of call paths to library $l_i$ as $n_i$. The final vector representation of third-party library usage is the vector of these raw counts, $f_{\text{lib}} = \{n_1, n_2, ..., n_k\}$. The subsequent layers of the neural network are responsible for learning to handle the scale and distribution of these features.

### 3.5 Model Architecture and Training

With the three feature vectors extracted, we train a model to fuse them and classify app behavior. The overview of our model is shown in Figure 3.

**Model Design.** The final model is a Multi-Layer Perceptron (MLP) designed to fuse the three heterogeneous feature vectors. The model takes $f_{\text{bin}}$, $f_{\text{cxt}}$, and $f_{\text{lib}}$ as inputs. Each vector is first projected to a common dimension via a dedicated fully-connected layer with a ReLU activation:

$$f'_{\text{view}} = \text{ReLU}(W_{\text{view}} f_{\text{view}} + B_{\text{view}}) \tag{1}$$

The three projected vectors are then concatenated to form the final input feature vector for the MLP classifier:

$$f = \text{Concat}(f'_{\text{bin}}, f'_{\text{cxt}}, f'_{\text{lib}}) \tag{2}$$

This concatenated vector $f$ is then passed through an MLP consisting of three hidden fully-connected layers. At each layer $i$, the feature vector is transformed as follows:

$$f_i = \text{ReLU}(W_i f_{i-1} + B_i) \tag{3}$$

**Undesired Behavior Classification.** The last layer of the model is the classification layer, which uses a *Softmax* activation function to produce a probability distribution over the target classes. We train the model end-to-end by minimizing a *categorical cross-entropy* loss function. This fusion architecture is designed for robustness: if one view is compromised (e.g., by code obfuscation affecting $f_{\text{bin}}$), the model can leverage the strong, independent signals from the other two views to make a correct prediction.

## 4 Evaluation

We evaluate the effectiveness of BINCTX on real-world malware and other apps that contain undesired behaviors. We aim to answer the following research questions:

- RQ1: How effective is BINCTX for malware-family and undesired-behavior classification?
- RQ2: How does BINCTX perform in the two tasks compared with the SOTA approaches?
- RQ3: How do different features affect the effectiveness of BINCTX?
- RQ4: How robust is BINCTX against code obfuscation and adversarial attacks?

### 4.1 Experimental Setup

**Datasets and Pre-processing.** Our evaluation is conducted on a comprehensive dataset curated from two primary sources to ensure representation of diverse malicious and undesired behaviors. The first is a malware family dataset derived from the Android Malware Dataset Wei et al. (2017), which includes 6 distinct malware families, each with over 1,000 samples. This is supplemented with 7,000 benign applications sourced from Google Play, each verified as clean by VirusTotal VirusTotal (2022). The second source is an undesired behavior dataset from prior work Hu et al. (2021); Wang et al. (2018), containing 2,992 real-world apps whose labels are based on user comments from a third-party platform KuChuan (2022). From this source, we filtered the data to focus on significant issues, discarding categories with fewer than 30 samples and those related to device-specific performance problems.

Table 1: Final Dataset Composition

| Category Type | Class Name | # Apps |
|---|---|---|
| *Malware Families Wei et al. (2017)* | | |
| | FakeInst | 1,199 |
| | Fusob | 1,199 |
| | Mecor | 1,200 |
| *Undesired Behaviors Hu et al. (2021); Wang et al. (2018)* | | |
| | Ad Disruption | 4,136 |
| | Payment Deception | 289 |
| | Illegal Redirection | 156 |
| *Benign* | (Verified clean) | 8,632 |

To create a unified classification task, we merged these two sources and re-labeled the samples based on their primary characteristics. Our final dataset consists of three prominent malware families (FakeInst, Fusob, Mecor), three critical undesired behaviors, and a consolidated benign class, with the final distribution detailed in Table 1. A crucial characteristic of this dataset is the high prevalence of code obfuscation; our analysis revealed that over 87% of benign apps and 96% of malicious/undesired apps employ obfuscation, underscoring the necessity for robust detection models.

**Implementation Details and Hyperparameters.** For our model, all input bytecode images were resized and padded to a uniform shape of $300 \times 300 \times 3$. The MLP classifier was designed with 3 hidden layers, each containing 3,000 neurons. We trained the model using the Adam optimizer with a batch size of 256. The evaluation was conducted using a standard 80% training and 20% testing split, with a 10-fold cross-validation protocol. We report precision, recall, and $F_1$-score as our primary performance metrics.

**Baselines for Comparison.** We compare BINCTX against five state-of-the-art baselines that represent three major methodological categories:

- Bytecode Embedding Approaches: CODEIMAGE Sun et al. (2019) and DEXIMAGE Kang et al. (2020).
- Metadata/API-based Approaches: A3CM Qiu et al. (2019) and ANDMFC Türker & Can (2019).
- Code Semantics-based Approach: DEEPREFINER Xu et al. (2018).

## 4.2 RQ1: OVERALL PERFORMANCE

Table 2: Comparison of F1-Scores across all approaches. Our method, BINCTX, outperforms all baselines in every category. Best results are in **bold**.

| Category | BINCTX | CODEIMAGE | DEXIMAGE | ANDMFC | A3CM | DEEPREFINER |
|---|---|---|---|---|---|---|
| *Benign* | **97.49** | 85.02 | 86.13 | 61.52 | 71.89 | 93.58 |
| Ad Disruption | **91.54** | 62.52 | 77.46 | 36.21 | 47.66 | 74.72 |
| Payment Deception | **89.87** | 42.94 | 54.41 | 33.85 | 37.06 | 67.73 |
| Illegal Redirection | **87.27** | 48.89 | 62.90 | 29.97 | 39.30 | 71.47 |
| FakeInst | **98.99** | 87.49 | 87.46 | 71.23 | 67.48 | 94.74 |
| Fusob | **99.50** | 89.20 | 89.10 | 69.67 | 61.82 | 84.73 |
| Mecor | **98.48** | 80.74 | 73.68 | 64.36 | 64.52 | 89.80 |
| **Average** | **94.73** | 70.97 | 76.06 | 51.99 | 55.83 | 82.43 |

Our primary results demonstrate that BINCTX is highly effective at detecting both malware and other undesired behaviors. As shown in Table 2, our model achieves a strong macro-average $F_1$-score of 94.73%. Performance is particularly strong on malware family classification (e.g., FakeInst and Fusob), which is attributable to the high intra-class similarity within malware families where most samples are variants of existing ones. Even for the more diverse undesired behavior categories, BINCTX's multi-modal approach maintains robust performance, achieving an average $F_1$-score of 89.56%.

**Qualitative and Error Analysis.** A qualitative review confirms the model's behavior. For malware, the near-identical bytecode images of samples within the same family provide a powerful visual signature. For undesired behaviors, other modalities are more decisive; for instance, apps flagged for Ad Disruption use ad library APIs over 60% more frequently than benign apps. Our error analysis reveals some class confusion between malware families with similar behaviors (e.g., Fusob and Mecor) and identifies evidence of potential label noise in the user-comment-driven dataset. As shown in Figure 4, some samples misclassified as benign are visually indistinguishable from correctly labeled benign apps, suggesting the subjectivity of the original labels.

### 4.3 RQ2: COMPARISON WITH SOTA APPROACHES

As shown in Table 2, BINCTX consistently outperforms all baselines, achieving a significant $F_1$-score improvement of at least 14.92% over the strongest competitor, DEEPREFINER. The performance gap is even more pronounced against other approaches. The baselines' failures stem from their limited feature representations, which struggle to capture the nuanced characteristics of undesired behaviors.

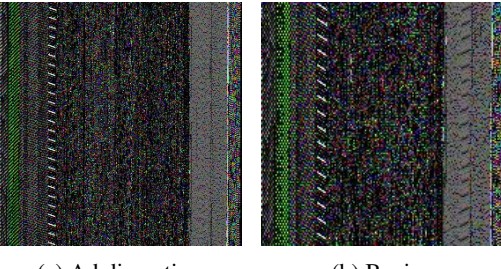

(a) Ad disruption      (b) Benign

Figure 4: DEX-as-image: ad disruption vs. benign.

The analysis reveals three key failure patterns. Bytecode-only models like CODEIMAGE and DEXIMAGE lack the contextual grounding to disambiguate behaviors that have similar visual code patterns but different intents. API-based approaches (ANDMFC, A3CM) rely on a narrow feature space of sensitive permissions and calls, rendering them ineffective against threats that intentionally avoid these signals. For instance, while FakeInst uses SMS permissions maliciously, many benign apps use them legitimately, making the signal unreliable. Finally, even the strongest baseline, DEEPREFINER, exhibits a critical failure mode: its XML-based first layer overfits to patterns in repackaged malware but fails on diverse undesired behaviors, while its opcode-based second layer lacks the necessary high-level context for accurate classification.

### 4.4 RQ3: ABLATION AND FEATURE IMPORTANCE ANALYSIS

To understand the contribution of each modality within our multi-modal framework, we conducted two key analyses: a direct ablation study comparing against a single-modality baseline, and a permutation feature importance test to quantify the relative influence of each view.

First, we performed an ablation study by creating a "bytecode-only" variant of our model, BINCTX$_{bc}$, which excludes the contextual and third-party library features. The results, shown in Table 3, are revealing. While BINCTX$_{bc}$ achieves comparable performance to the full BINCTX model on malware families with strong visual signatures, its $F_1$-score drops by at least 6.71% on the more diverse undesired behavior categories. Overall, the full BINCTX model achieves a 24% higher area under the ROC curve (AUC) score (0.9068 vs 0.7288). This confirms that while the visual representation is a powerful foundation, the contextual and library usage features are indispensable for accurately classifying nuanced, non-traditional threats. Notably, even as a single-modality model, BINCTX$_{bc}$ still outperforms other bytecode-only baselines like CODEIMAGE and DEXIMAGE, which we attribute to its more advanced DenseNet-based image embedder.

Table 3: Detailed performance of BINCTX$_{bc}$.

| Category | Prec. (%) | Rec. (%) | F1 (%) |
|---|---|---|---|
| *Benign* | 95.64 | 94.10 | 94.86 |
| Ad Disruption | 82.20 | 89.47 | 85.78 |
| Payment Deception | 60.86 | 58.33 | 59.67 |
| Illegal Redirection | 64.53 | 61.92 | 63.20 |
| FakeInst | 96.00 | 100.00 | 97.96 |
| Fusob | 99.00 | 100.00 | 99.50 |
| Mecor | 97.00 | 100.00 | 98.48 |
| **Average** | **85.03** | **86.26** | **85.64** |

Table 4: Feature importance of BINCTX.

| Feature | Value |
|---|---|
| Bytecode Representation | 0.58 |
| Third-Party Library Usage Pattern | 0.24 |
| Contextual Feature | 0.09 |

To further quantify the relative importance of each modality, we employed a permutation feature importance analysis. This technique measures the drop in model accuracy when a single feature view is randomly shuffled, thereby breaking its correlation with the target label. The results, summarized in Table 4, indicate that the Bytecode Representation is the most critical feature, with an importance value of 0.58. It is followed by the Third-Party Library Usage Pattern at 0.24, and the Contextual Feature view at 0.09. These results validate our multi-modal design, confirming that all three views provide a positive and significant contribution to the model's predictive power.

## 4.5 RQ4: ROBUSTNESS ANALYSIS

A critical requirement for a practical detection model is its robustness against common evasion techniques. We evaluate BINCTX's resilience against two primary threats: commercial-grade code obfuscation and adversarial attacks.

**Resistance to Code Obfuscation.** Our main evaluation dataset already reflects real-world conditions, with over 80% of samples employing obfuscation, yet BINCTX achieves high precision. To further and more rigorously test this capability, we conducted a controlled experiment on 400 open-source apps from F-Droid fdr (2023). On the original clean apps, our trained model achieved a 90% $F_1$-score. After applying a commercial reinforcement framework jia (2023) to heavily obfuscate these apps, the same trained model showed only a modest performance drop, achieving a remarkable 84% $F_1$-score. This graceful degradation demonstrates that BINCTX's multi-modal representation does not rely on brittle, superficial code patterns.

**Resistance to Adversarial Attacks.**

We also tested BINCTX's resilience against adversarial samples from prior work Li & Li (2020), which were generated with a mixture of attacks including dead code injection and manifest manipulation. As shown in Table 5, BINCTX again demonstrates superior robustness, outperforming the strongest baseline by over 17.60% in $F_1$-score. The analysis reveals why: while the visual representations of bytecode-only models are brittle against dead code injection and DEEPREFINER's XML features are susceptible to manifest manipulation, BINCTX's robustness stems directly from its multi-modal design. Its analysis of contextual and library features is guided by the app's ICCG from active entry points, naturally ignoring unreachable injected code and providing a stable signal for classification. This highlights a key architectural advantage of fusing a global visual representation with semantics derived from reachable code paths.

Table 5: Effectiveness on adversarial samples

| Approach | Prec. (%) | Rec. (%) | F1 (%) |
|---|---|---|---|
| BINCTX | 81.79 | 73.05 | 77.17 |
| CODEIMAGE | 46.63 | 40.90 | 43.57 |
| DEXIMAGE | 47.75 | 41.48 | 44.39 |
| DEEPREFINER | 60.93 | 58.27 | 59.57 |

## 5 DISCUSSION

Our choice to encode DEX bytes as RGB images is a pragmatic trade-off, retaining local byte neighborhoods for texture analysis more effectively than grayscale alternatives. A key limitation, however, is the fixed $300 \times 300 \times 3$ input size, which can cause information loss for larger applications via resizing. While our current approach yields strong results, future work could explore multi-scale or patch-based encoders to improve feature fidelity.

The static analysis for SDK usage, while effective, also has inherent limitations. Our ICCG-based path counting can over-approximate reachability and does not capture runtime call frequencies. Furthermore, our static view is vulnerable to advanced evasion techniques like reflection, dynamic code loading, and native code. Although our multi-modal features provide some resilience, as shown in Section 4.5, explicitly integrating recent de-reflection and native-code analysis techniques Sun et al. (2021); Samhi et al. (2022); Wei et al. (2018) is a promising direction for future work.

Finally, our evaluation scope is subject to practical constraints. Our malware dataset, while large, does not cover all known families, particularly those with few samples. The labels for undesired behaviors, being derived from user comments, are inherently subjective despite our filtering efforts. We also intentionally excluded performance-related issues that are ill-suited for static analysis. Future work could strengthen the external validity of our findings by incorporating ground-truth data from industry partners, using stronger label sources, and automating the identification of third-party SDKs.

## 6 RELATED WORK

**Android Malware Classification.** Deep learning has been widely applied to Android malware detection, using features such as opcode sequences, dangerous API calls, and string values Xu et al. (2018); McLaughlin et al. (2017); Kim et al. (2019); Zhang et al. (2014); de la Puerta et al. (2015); Santos et al. (2013); Hou et al. (2016); Maiorca et al. (2017). However, these approaches often have two key limitations. First, their chosen features can be brittle and susceptible to simple evasion techniques like dead code insertion. Second, they are frequently trained on well-known but older datasets like AMD Wei et al. (2017) and Drebin Arp et al. (2014), which are dominated by repackaged malware. These datasets do not adequately represent the diversity of modern, non-traditional undesired behaviors, limiting the generalizability of models trained on them.

**Undesired Behaviors.** Prior work has also focused specifically on undesired behaviors. DeepIntent Xi et al. (2019), for example, detects discrepancies between UI icons and the sensitive permissions they trigger, but its scope is limited to behaviors involving such permissions. CHAMP Hu et al. (2021) leverages user comments to characterize and categorize policy violations like aggressive advertising. While valuable for understanding the problem space, it is not a direct detection model. In contrast, our work provides a direct, learning-based detector that is not reliant on dangerous permissions.

**Code Embedding.** Our work relates to the broader field of code embedding. While sophisticated models based on Abstract Syntax Trees (ASTs) or Graph Neural Networks (GNNs) have shown promise for representing source code Alon et al. (2019); Chen et al. (2018); Zhang et al. (2019b); Zhao & Huang (2018); Zhou et al. (2019), their high computational complexity makes them challenging to apply to entire, large-scale Android applications. Furthermore, building the precise program graphs they require is difficult in the event-driven Android environment. Another line of work uses hash-based embeddings to find syntactically similar code Indyk & Motwani (1998); Datar et al. (2004); Weiss et al. (2008). While useful for detecting repackaged apps, this approach is ineffective for identifying semantically similar but independently implemented undesired behaviors. Our lightweight bytecode-as-image approach bypasses these limitations by providing a holistic and efficient representation.

## 7 CONCLUSION

We have presented BINCTX, a novel approach that extracts image representation of bytecode and contextual features such as the usage patterns of third-party libraries and URL constants build a general behavior model. As the image representation shows the global overview of apps' behaviors and contextual features capture specific patterns of undesired behaviors such as the uses of third-party libraries, our model is effective in classifying not only malicious behaviors into malware families, but also undesired behaviors that do not request dangerous permissions. Our evaluations on real-world malware and apps that contain undesired behaviors demonstrate the effectiveness of BINCTX in detecting both undesired and malicious behaviors. Furthermore, we also show that BINCTX is more resistant to adversarial samples than the baseline approaches.

REPRODUCIBILITY.

To ensure the reproducibility of our work, we provide a comprehensive set of artifacts. Our submission includes the anonymized source code for our data pipelines and model, along with scripts to replicate our experiments. The necessary environment specifications, including all software dependencies and their versions, are also documented. This documentation, in conjunction with the supplied scripts, allows for the full reproduction of all metrics and tables reported in this paper.

THE USE OF LARGE LANGUAGE MODELS

We used a large language model solely to aid in writing polish. Specifically, it was employed for:

1. Improving grammar, clarity, and conciseness of paragraphs;

2. Suggesting alternative phrasings to enhance readability;

3. Typesetting optimization and figure/table formatting to improve the clarity of result presentation.

No model outputs were used for designing experiments, analyzing results, generating figures/tables, or providing substantive research contributions. All ideas, methods, experiments, and analyses were entirely conducted by the authors.

ACKNOWLEDGMENTS

We thank anonymous reviewers for constructive feedback. This work was supported in part by institutional grants and industry collaborators.

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
