# OpenReview forum: "BinCtx: Multi-Modal Representation Learning for Robust Android App Behavior Detection"
_ICLR.cc/2026/Conference — ICLR 2026 Conference Withdrawn Submission_

### Official Review · Reviewer_3YWn · 2025-10-28

**Soundness:** 3
**Presentation:** 2
**Contribution:** 2
**Rating:** 2
**Confidence:** 5

**Summary:**

BINCTX targets both Android malware and market-policy “undesired behaviors” by learning a tri-view, multi-modal representation per APK. It fuses a DenseNet-based bytecode-as-image view, a contextual view, and third-party SDK usage derived from ICCG paths; the three embeddings are concatenated for classification. Experiments span benign apps, policy-violating behaviors, and malware families, reporting macro-F1 up to 94.73% over strong bytecode-only and metadata/API baselines. Stress tests show a modest drop from 90% to 84% F1 under commercial obfuscation and >17.6% macro-F1 gains over prior adversarial samples. While the system is well-organized and empirically competitive, the problem framing blends malware detection with policy-driven categories, and evidence for cross-market/temporal generalization and comparisons to stronger graph/sequence models is limited, complicating assessment of originality and robustness.

**Strengths:**

1.	The tri-view, multi-modal design—bytecode-as-image, contextual manifest/URL features, and third-party SDK path usage—creatively fuses complementary signals to cover malware and policy-violating “undesired behaviors.”
2.	Strong empirical results: macro-F1 94.73% overall, with large gains over SOTA baselines; clear ablations and permutation importance quantify each view’s contribution.
3.	The system pipeline is well-structured (Figures/Sections on extraction and fusion) with explicit feature definitions and model equations, enabling reproducibility.
4.	Demonstrated robustness under commercial obfuscation (F1 ≈84%) and adversarial samples (+17.6% F1 over best baseline), addressing real-world evasion.

**Weaknesses:**

Unclear motivation and architectural rationale: The paper’s multimodal fusion design lacks a convincing motivation-to-method linkage. The combination of bytecode, manifest context, and SDK usage appears heuristic rather than necessity-driven, giving the impression of a feature-stacking system instead of a conceptually grounded representation framework.

Dataset composition and label reliability: Samples are drawn from heterogeneous and temporally inconsistent sources. Malware family data and undesired-behavior data stem from different periods and labeling pipelines (e.g., VirusTotal engines vs. user-comment-based annotations), which may reflect incompatible labeling criteria. Such heterogeneity could bias training and inflate reported robustness.

Outdated or incomplete baselines: The comparative study mainly includes older bytecode or metadata-based systems, omitting recent multimodal and graph-sequence hybrid methods (e.g., MalRadar 2022, Mitigating Emergent Malware Label Noise in DNN-Based Android Malware Detection 2025). This undermines the credibility of the claimed performance margin and limits the paper’s novelty and relevance.

**Questions:**

Motivation clarity: The current description makes the three-modal design appear empirical rather than necessity-driven. Could the authors explicitly map each modality to the concrete detection challenges identified in Section 1? For example, how does the bytecode view specifically mitigate obfuscation or opcode noise, how does the contextual manifest view capture trigger semantics or background activation behaviors, and how does the library-usage view characterize SDK-driven commercial deception? A clearer causal linkage from challenge → modality → fusion rationale would greatly strengthen the conceptual foundation and dispel the impression of feature concatenation.

Dataset validity and consistency: The dataset merges samples from distinct origins and timeframes—malware families labeled by anti-virus engines and undesired behaviors derived from user comments or market metadata. How are temporal drifts, labeling schema discrepancies, and engine-specific taxonomies reconciled? Were any normalization, re-annotation, or cross-verification procedures performed to ensure that family and behavior labels are semantically comparable? Otherwise, the mixture may encode dataset-specific biases, calling into question the robustness and reproducibility of the reported F1 gains.

Baseline adequacy and relevance: The evaluation omits several recent deep graph, sequence, and transformer-based malware detectors (e.g., MalRadar 2022, Mitigating Emergent Malware Label Noise in DNN-Based Android Malware Detection 2025). What criteria guided baseline selection? Could the authors include or at least discuss comparisons with these more advanced multimodal systems, and analyze the margin under identical training splits? Without this, the claimed “state-of-the-art” status remains difficult to substantiate.

---

### Official Review · Reviewer_B3h2 · 2025-10-29

**Soundness:** 2
**Presentation:** 3
**Contribution:** 2
**Rating:** 4
**Confidence:** 4

**Summary:**

The authors propose ​​BINCTX​​, a robust Android malware detection model that integrates three distinct feature modalities. These features are embedded and fused to train a contextual-aware classifier. Experimental results demonstrate that BINCTX achieves superior performance under both obfuscation and adversarial attack scenarios.

**Strengths:**

The paper is well-structured and addresses an important research problem.

**Weaknesses:**

The ​​novelty and motivation​​ require significant clarification. This work does not sufficiently establish why the fusion of these three particular featuresis a novel and well-motivated solution. A deeper discussion is needed to explain the theoretical basis for their complementarity.

**Questions:**

1.The concept of robustness mentioned in the title of the paper is not clearly defined in the introduction, which leads to ambiguity regarding the core contributions of the research. Does the robustness referred to in the title pertain to resilience against adversarial attacks and obfuscation? If so, what are the issues with the current features being used, and why were these three particular features chosen? Additionally, since different obfuscation techniques affect various features differently, how do these obfuscation methods impact different features in distinct ways?

2.The motivation behind the study needs further elaboration. Merely stating what information image features encompass is insufficient, as many detection models currently utilize image features. For instance, it would be beneficial to explain why the selected combination of features offers complementary information, which in turn contributes to their effectiveness in resisting obfuscation.

3.In section 3.2, the author claims that image representation is more effective compared to CFG and FCG. It would be beneficial to clarify why it is more efficient. Moreover, if the primary advantage is merely efficiency, the author's use of a variety of features integrated into a single model inherently increases the time required for feature extraction. This seems to contradict the initial motivation presented.

4.The ablation study needs to be more detailed, for example, by examining the individual effects of the three different features and the effects of their pairwise combinations.

5.Regarding robustness, some studies have focused on detecting adversarial examples or obfuscated APKs. The author should include comparisons with these existing methods.

6.An assessment of detection efficiency is necessary. The time taken by each module should be statistically analyzed.

---

### Official Review · Reviewer_cs2v · 2025-11-04

**Soundness:** 2
**Presentation:** 2
**Contribution:** 2
**Rating:** 2
**Confidence:** 4

**Summary:**

This paper proposes a multimodal representation learning method BINCTX to detect malicious and undesirable behaviors in Android applications. The method fuses three features to train a multilayer perceptron classifier. Experiments show that BINCTX is better than existing methods and shows strong robustness against code obfuscation and sample attacks.

**Strengths:**

+ The article is well written and clearly structured.
+ Multi-modal structures enhance resistance to attacks.

**Weaknesses:**

+ The components in the proposed method have been discussed in the previous work, and it is not new to convert binary code into images.
+ Experimental settings are not well described. Lack of necessary implementation details, such as the dimensions of embedding and the specific conversion method from bytecode to pixel.
+ The comparative literature is old and lacks relevant work in the past three years.

**Questions:**

+ An Android software may be split into multiple APKs. Are the tasks performed apk-level detection or software software-level detection?
+ An APK contains a large number of bytecodes. Why choose the current image size? How to convert a particularly large APK file into an image representation?
+ Why consider converting bytecode to an image, and what advantages does this have compared to using the original sequence?
+ What third-party libraries were considered and why? What is the dimension of one-hot embedding?
+ CODEIMGAE and DEXIMAGE also use the method of constructing images for representation learning. Why does BINCTX_BC achieve better results?
+ One of the current challenges is 2. Insufficient modeling of behavioral context, however, as shown in Table 4, the feature importance of contextual features is 0.09. What is the actual contribution of these features to performance?
+ Can the proposed method detect both malware and undesirable behavior? Is this a four-category model or a seven-category model? Figure 3 shows five categories, which is confusing. I guess it's four categories; if so, how is the performance of the benign category in Table 2 evaluated?

---

### Note · Authors · 2025-11-20

I have read and agree with the venue's withdrawal policy on behalf of myself and my co-authors.